

# Dilaton in scalar QFT: A no-go theorem
# in $4-\varepsilon$ and $3-\varepsilon$ dimensions

**Daniel Nogradi and Balint Ozsvath**

Eotvos University, Department of Theoretical Physics
Pazmany Peter setany 1/a, Budapest 1117, Hungary

## Abstract

Spontaneous scale invariance breaking and the associated Goldstone boson, the dilaton, is investigated in renormalizable, unitary, interacting non-supersymmetric scalar field theories in $4-\varepsilon$ dimensions. At leading order it is possible to construct models which give rise to spontaneous scale invariance breaking classically and indeed a massless dilaton can be identified. Beyond leading order, in order to have no anomalous scale symmetry breaking in QFT, the models need to be defined at a Wilson-Fisher fixed point with exact conformal symmetry. It is shown that this requirement on the couplings is incompatible with having the type of flat direction which would be necessary for an exactly massless dilaton. As a result spontaneous scale symmetry breaking and an exactly massless dilaton can not occur in renormalizable, unitary $4-\varepsilon$ dimensional scalar QFT. The arguments apply to $\phi^6$ theory in $3-\varepsilon$ dimensions as well.


# 1 Introduction

A massless (or approximately massless) dilaton is thought to arise in a number of field theories where (approximate) scale invariance is spontaneously broken. If this happens an effective field theory should be able to describe the low energy dynamics of the dilaton along with other potentially light degrees of freedom. These other light degrees of freedom are often also Goldstone bosons originating from spontaneous breaking of other symmetries besides scale invariance. These other symmetries might be also approximate and in this case the corresponding Goldstones would also be only approximately massless.

Conformal invariance, or invariance only by a subgroup, scale transformations, may play a role in the Standard Model and its extensions in a number of ways. The only scale symmetry breaking parameter of the Standard Model is the Higgs mass, at least at leading order. Quantum effects lead to additional scale symmetry breaking (dimensionful parameters), most notably by anomalous breaking of scale invariance and by spontaneous breaking of global symmetries. One class of ideas explores the possibility that scale symmetry itself is spontaneously broken and to what extent the Higgs boson (or other light particles) may be identified with the corresponding light dilaton [1–12]. In particular, the smallness of the ratio between the Higgs mass and the Planck scale, which appears to be an enormously fine tuned quantity, is then explained by scale symmetry and its spontaneous breaking [13]. The same could be said about another fine tuning problem of the Standard Model, the smallness of the cosmological constant as well [14]. For early appearances of spontaneous breaking of scale invariance and its implications in QFT see [15–17].

The Higgs boson may also be viewed as a generic composite particle in several extensions of the Standard Model inspired by strong dynamics [18, 19] with or without a dilatonic interpretation. In these scenarios the spectrum of a strongly interacting new sector is thought to include a composite light scalar and if so, may be identified with the Higgs. Furthermore, the lightness may be related to the dilatonic nature of the particle although this interpretation is far from clear [20, 21]. In any case, the strongly interacting nature of the underlying theory makes the study of detailed properties of the low energy excitations difficult, and served as the major motivation for a surge in non-perturbative studies recently [22–37] as well as descriptions in terms of effective theories [38–40].

Whether a particular light degree of freedom in a given field theory can be identified as a dilaton is often non-trivial. Even though the effective theory might be weakly coupled, the underlying theory is often strongly coupled, complicating the identification of the physical degrees of freedom between the two. The fact that the light particles in question, one of which would be the hypothetical dilaton, are often not exactly massless further complicates the precise identification, especially if there are no parameters which would control the masses separately. For example, non-abelian gauge theories are often thought to give rise to a light dilaton if the fermion content is not far from the conformal window, but the lack of an obvious control parameter for its mass makes this conclusion more conjectural than firmly established. In this example the underlying theory is strongly coupled so the bridge from it to the effective theory is beyond analytical tools. Hence it is not possible to simply derive the effective theory in a top-down approach. As is often the case with effective theories one needs to consider the most general model with the given degrees of freedom and symmetries and match the unknown coefficients to observables in the underlying theory. However, if it is not known whether the underlying theory does give rise to a dilaton or not, one will not know whether a dilaton field should be included in the effective theory to begin with. Even if a dilaton is included in the effective theory, its interactions are not sufficiently constrained by scale symmetry. In contrast, the effective theory describing Goldstones corresponding to chiral symmetry breaking in gauge theory is essentially unique and fixed by the pattern of symmetry breaking. If a dilaton is to

be coupled to these Goldstones, the form of the coupling is not fixed by scale symmetry and there are different conjectures for the detailed form of the coupled system.

The main motivation for the present work was to construct a renormalizable, weakly coupled and unitary theory in which spontaneous scale symmetry breaking unambiguously takes place and is fully in the realm of perturbation theory. In this case it would be possible to follow the dynamics from the underlying theory down to the effective theory describing the massless dilaton and potentially the other Goldstones, in a fully controlled manner. In order to avoid special features specific only to supersymmetric theories, non-supersymmetric models are sought.

Ideally, one would start with a weakly coupled CFT with a vanishing $\beta$-function and break scale symmetry spontaneously only, generating massive particles as well as the massless dilaton. We know of no renormalizable, non-supersymmetric and unitary such example in 4 dimensions [9]. Hence we will be working in $4-\varepsilon$ dimensions and purely scalar QFT for simplicity, where the Wilson-Fisher fixed points [41] provide the necessary starting point as a weakly coupled CFT.

The main result is that the two essential features of the interaction are incompatible, namely that (1) spontaneous scale symmetry breaking is present (2) the model is renormalizable in the UV and an IR fixed point also exists, can not be met simultaneously. Hence in renormalizable unitary scalar QFT in $4-\varepsilon$ dimensions spontaneous scale symmetry breaking can not take place and consequently a massless dilaton can not exist.

It is well known that supersymmetric models with the desired properties do exist, even in 4 dimensions. Most notable are $\mathcal{N} = 2, 4$ SUSY Yang-Mills [42]. If the scalar vevs are all zero, scale symmetry (even the larger full conformal group) is intact and in QFT all particles are massless with a vanishing $\beta$-function. Giving non-zero vevs to the scalars leads to spontaneous scale symmetry breaking and a corresponding exactly massless dilaton, along with other massless and massive states. Supersymmetry ensures that the flat direction required for the dilaton is not lifted by quantum corrections at any order; see e.g. [43].

Recently, the first non-supersymmetric 4 dimensional example was found [44]. In this example the double scaling limit of $\gamma$-deformed $\mathcal{N} = 4$ SUSY Yang-Mills [45–48] is considered at strong deformation and weak coupling, in the large-$N$ limit, leading to so-called fishnet CFT's [44, 49]. The resulting theory is purely bosonic and renormalizable, however it is non-unitary. Non-unitarity is in fact crucial to show that the flat direction giving rise to a massless dilaton is not lifted by quantum effects.

These observations about supersymmetric and non-unitary non-supersymmetric examples were our primary motivation for the present work. Our result shows that a well-defined dilaton in renormalizable, non-supersymmetric and unitary perturbative QFT is not easy to construct; in purely scalar QFT it is in fact impossible in $4-\varepsilon$ dimensions. For a recent review on the fate of scale symmetry in QFT along with phenomenological applications in cosmology, gravity and particle physics in general, see [14].

The interaction in $4-\varepsilon$ dimensions is of course quartic. The arguments given in this case can be applied to $\phi^6$ theory in $3-\varepsilon$ dimensions as well. The leading order 2-loop $\beta$-function has a zero at small $O(\varepsilon)$ coupling, just as with the Wilson-Fisher fixed point in $4-\varepsilon$ dimensions. The couplings at the fixed points correspond to a potential which does not have a flat direction necessary for spontaneous scale symmetry breaking and hence a massless dilaton.

In section 2 a number of examples are provided at leading order demonstrating that a classical dilaton can easily arise, either as the sole massless mode or together with other Goldstone bosons. Section 3 deals with the 1-loop $\beta$-functions of each example and is shown that an IR fixed point can not be reached in the available space of couplings. The general argument is given in section 4 why the presence of an IR fixed point precludes the scalar potential from having a flat direction necessary for spontaneous scale symmetry breaking and a massless dila-

ton in $4-\varepsilon$ dimensions. In section 5 the same is shown in $3-\varepsilon$ dimensions with $\phi^6$ theory. We end with a set of conclusions and future outlook in section 6.

## 2  Dilaton at leading order in 4 dimensions

We are seeking interacting renormalizable scalar field theories described by scale invariant actions with the property that this symmetry breaks spontaneously only. Classically, there is no problem with working in 4 dimensions for illustrating the tree-level structure. The generic form at tree level is,

$$\mathscr{L} = \frac{1}{2}\partial_\mu \phi_a \partial^\mu \phi_a - \mathscr{V}(\phi), \tag{1}$$

where the potential $\mathscr{V}$ contains only dimensionless couplings. No particular global symmetry is imposed, only scale symmetry, leading to

$$\mathscr{V}(\phi) = \frac{\lambda_{abcd}}{4!}\phi_a \phi_b \phi_c \phi_c. \tag{2}$$

The real couplings $\lambda_{abcd}$ and real fields $\phi_a$ are understood to be bare quantities. Scale invariance is clearly present, the action $S = \int d^4x \mathscr{L}$ is invariant under the space-time symmetry $x \to e^{-s}x$ once the scalar field $\phi_a$ is transformed according to its mass dimension, $\phi_a(x) \to e^s \phi_a(e^s x)$. In particular linear, quadratic or cubic terms in the fields are not allowed.

The potential $\mathscr{V}$ is required to be non-negative for stability of the vacuum. Then clearly $\phi_a = 0$ always corresponds to a vacuum state, one which does not break scale invariance and all particles are massless. We would like to construct potentials which possess non-trivial minima $\phi_a = v_a \neq 0$. Once such a minimum exists the corresponding vacuum will break scale invariance and some states will acquire masses proportional to $v_a$.

There is no obstruction at leading order, the simplest example is given by a two component model,

$$\mathscr{V} = \frac{\lambda}{4}\phi_1^2 \phi_2^2. \tag{3}$$

Clearly the potential is non-negative and possesses infinitely many stable minima. The choice $(\phi_1, \phi_2) = (0, 0)$ corresponds to a scale symmetric vacuum, while $(\phi_1, \phi_2) = (v, 0)$ and $(\phi_1, \phi_2) = (0, v)$, with arbitrary $v$, correspond to vacua which break scale invariance spontaneously.

Expanding around the scale invariant vacuum leads to two massless bosons interacting through a quartic interaction. More interesting is the expansion around a scale symmetry breaking vacuum, for definiteness let us choose $(\phi_1, \phi_2) = (0, v)$, and the corresponding fluctuating fields will be denoted by $\eta$ and $\chi$,

$$\begin{aligned}\phi_1 &= \eta, \\ \phi_2 &= v + \chi.\end{aligned} \tag{4}$$

The potential becomes,

$$\mathscr{V} = \frac{\lambda}{4}v^2 \eta^2 + \frac{\lambda}{2}v \eta^2 \chi + \frac{\lambda}{4}\eta^2 \chi^2, \tag{5}$$

which clearly describes a massive particle $\eta$ with $M^2 = \frac{1}{2}\lambda v^2$ and a massless particle $\chi$, the dilaton [1]. The two types of particles are interacting through a cubic and quartic interaction.

---

[1]In this particular example a global $\mathbb{Z}_2 \times \mathbb{Z}_2$ given by flipping the sign of the two fields is also broken to $\mathbb{Z}_2$ but this is of no significance for our discussion. Same applies to the models (6) and (9).

Goldstone's theorem applies, the direction given by $\chi$ is a flat direction of the potential and hence corresponds to a massless particle. It is well-known that spontaneous breaking of space-time symmetries behave differently from spontaneous breaking of global symmetries in terms of counting Goldstone bosons, both in non-Lorentz invariant [50] and in Lorentz invariant theories [51]. In the case of scale symmetry the naive counting however does apply, one spontaneously broken symmetry corresponds to one Goldstone boson.

It is possible to generalize the model (3) to describe an interacting $n$-component and a 1-component field, $\phi_a$ and $\Phi$. The potential

$$\mathcal{V} = \frac{h_{abcd}}{4!}\phi_a\phi_b\phi_c\phi_d + \frac{1}{2}g_{ab}\phi_a\phi_b\Phi^2, \tag{6}$$

with dimensionless couplings $h_{abcd}$ and $g_{ab}$ fulfilling suitable stability conditions gives rise to a scale symmetry respecting vacuum $(\phi_a,\Phi)=(0,0)$ as well as scale symmetry breaking ones $(\phi_a,\Phi)=(0,v)$ with arbitrary $v$. Expanding around one of these latter vacua we have fluctuating fields $\eta_a$ and $\chi$,

$$\begin{aligned} \phi_a &= \eta_a, \\ \Phi &= v + \chi, \end{aligned} \tag{7}$$

leading to the potential,

$$\mathcal{V} = \frac{1}{2}g_{ab}\eta_a\eta_b v^2 + g_{ab}v\eta_a\eta_b\chi + \frac{1}{2}g_{ab}\eta_a\eta_b\chi^2 + \frac{h_{abcd}}{4!}\eta_a\eta_b\eta_c\eta_d. \tag{8}$$

Before symmetry breaking we had $n+1$ massless particles. After scale symmetry breaking we have $n$ massive particles with mass matrix $M_{ab}^2 = g_{ab}v^2$ and a massless dilaton described by $\chi$ and again the two types of particles are interacting through cubic and quartic interactions. Goldstone's theorem applies again, one spontaneously broken symmetry corresponds to one massless Goldstone boson.

Something curious is nonetheless going on relative to generic field theories with Goldstone bosons. Generically, without symmetry breaking particles are massive and Goldstone's theorem provides an explanation why some particles *become* massless once the symmetry is spontaneously broken. In our case, since we start with a scale invariant action, all particles are massless from the start if scale symmetry is intact. Hence massless particles do not require a special explanation, their vanishing mass is simply a consequence of intact scale symmetry. After scale symmetry breaking a mass scale is generated and Goldstone's theorem ensures that one particle *remains* massless. At the same time all other particles acquire a mass. The non-trivial content of Goldstone's theorem in this case seems to be the precise number of particles which become *massive*, as opposed to becoming *massless*, after symmetry breaking.

As a third and final example let us incorporate spontaneous global symmetry breaking along with scale symmetry breaking, in which case we expect a dilaton as well as other Goldstone bosons. A potential with these properties is,

$$\mathcal{V} = \frac{\lambda}{4!}\left(\phi_a\phi_a - \Phi^2\right)^2. \tag{9}$$

The model is clearly scale invariant and also has an $O(n)$ symmetry. The trivial vacuum $(\Phi,\phi_a)=(0,0)$ again breaks neither scale invariance nor the global symmetry, while the non-trivial vacua,

$$(\Phi,\phi_1,\ldots,\phi_{n-1},\phi_n)=(v,0,\ldots,0,v), \tag{10}$$

with arbitrary $v$ does break both. In particular $O(n)$ breaks to $O(n-1)$ hence we expect $n$ Goldstones, $n-1$ from the breaking of the global symmetry and an additional one as the dilaton. Indeed, if fields $\eta_0, \eta_1, \ldots, \eta_n$ are introduced,

$$
\begin{aligned}
\Phi &= v + \eta_0, \\
\phi_A &= \eta_A, \qquad 1 \le A \le n-1, \\
\phi_n &= v + \eta_n,
\end{aligned}
\tag{11}
$$

the potential becomes

$$
\mathcal{V} = \frac{\lambda}{6} v^2 (\eta_n - \eta_0)^2 + \frac{\lambda}{6} v (\eta_n - \eta_0)(\eta_A^2 + \eta_n^2 - \eta_0^2) + \frac{\lambda}{4!}(\eta_A^2 + \eta_n^2 - \eta_0^2)^2.
\tag{12}
$$

A change of basis to $\xi = (\eta_n - \eta_0)/\sqrt{2}$ and $\chi = (\eta_n + \eta_0)/\sqrt{2}$ then leads to

$$
\mathcal{V} = \frac{\lambda}{3} v^2 \xi^2 + \frac{\lambda \sqrt{2}}{6} v \xi (\eta_A^2 + 2\xi\chi) + \frac{\lambda}{4!}(\eta_A^2 + 2\xi\chi)^2,
\tag{13}
$$

which shows that $\xi$ is massive with $M^2 = \frac{2}{3}\lambda v^2$ and the remaining $n$ particles are massless, $\chi$ is the dilaton and $\eta_A$ are the $n-1$ Goldstones corresponding to the breaking $O(n) \to O(n-1)$.

Goldstone's theorem applies again, out of the $n+1$ massless particles, exactly one becomes massive after symmetry breaking, $n$ remain massless.

Model (9) may be generalized further to the so-called biconical models. The field $\Phi$ in this case has $m$ components enlarging the original symmetry to $O(n) \times O(m)$. We will not discuss this setup further, rather just note that it has recently attracted interest; see [52] for a detailed discussion.

For completeness we note that the dilaton $\chi$ is often parametrized as $\chi = f e^{\sigma/f}$ with a new field $\sigma$ and dimensionful parameter $f$. Scale transformations $x \to e^{-s}x$ are realized non-linearly on $\sigma$,

$$
\sigma(x) \to \sigma(e^s x) + f s.
\tag{14}
$$

The discussion so far was completely classical and we turn to loop corrections in the next section.

## 3 Quantization in $4-\varepsilon$ dimensions

As shown in the previous section there is no obstruction to unambiguously define an exactly massless dilaton classically using a suitably chosen potential. One might wonder if a consistent renormalizable QFT can be built using the corresponding tree-level potentials. In particular we are seeking a non-trivial CFT with vanishing $\beta$-function. Were such a construction with spontaneous scale symmetry breaking successful, it would provide an example of a renormalizable interacting QFT describing some massive particles and a massless dilaton which is not just an effective theory. The main conclusion from this section will be that this is actually not possible with the examples given in section 2. In the next section it will be shown that the same conclusion applies generally.

Let us work within dimensional regularization and $\overline{\text{MS}}$ scheme and all couplings and fields will be assumed to be renormalized in this section. Since the starting point ought to be a weakly coupled CFT we work in $4-\varepsilon$ dimensions without anomalous scale symmetry breaking, which would occur in 4 dimensions. For small $\varepsilon$ the Wilson-Fisher IR fixed points, $\lambda_{abcd} = O(\varepsilon)$ are all perturbative. Once $\varepsilon > 0$ the potential in (1) of course picks up an additional $\mu^\varepsilon$ term, $\mathcal{V} \to \mu^\varepsilon \mathcal{V}$, for dimensional reasons.

First, let us see the effect of loop corrections on our simplest model (3). If we are to have the property that a non-trivial scale symmetry breaking vacuum exists in QFT, $\langle \phi_2 \rangle \neq 0$, the form of the potential should either remain the same as in (3) or only a term of the type $\phi_1^4$ should be generated. In other words the term $\phi_2^4$ is forbidden. Unfortunately there is no symmetry which would prohibit this term at all loop order, hence it is expected that it will be generated perturbatively. Indeed, if all terms are included at tree-level which are generated at 1-loop, so that the theory is renormalizable, we must start with the potential,

$$\mathscr{V} = \frac{\lambda_1}{4!}\phi_1^4 + \frac{\lambda_2}{4!}\phi_2^4 + \frac{\lambda_{12}}{4}\phi_1^2\phi_2^2. \tag{15}$$

The set of renormalized couplings $\lambda_1, \lambda_2, \lambda_{12}$ do close under the RG flow, at 1-loop we have,

$$\begin{aligned}
\mu\frac{d\lambda_1}{d\mu} &= -\varepsilon\lambda_1 + \frac{3}{16\pi^2}\left(\lambda_1^2 + \lambda_{12}^2\right), \\
\mu\frac{d\lambda_2}{d\mu} &= -\varepsilon\lambda_2 + \frac{3}{16\pi^2}\left(\lambda_2^2 + \lambda_{12}^2\right), \\
\mu\frac{d\lambda_{12}}{d\mu} &= -\varepsilon\lambda_{12} + \frac{1}{16\pi^2}\lambda_{12}\left(\lambda_1 + \lambda_2 + 4\lambda_{12}\right).
\end{aligned} \tag{16}$$

It is clear what the problem is: the subspace $\lambda_2 = 0$ is not invariant under the RG flow and there is no IR fixed point in this plane. The tree-level potential corresponding to $\lambda_2 = 0$ and $\lambda_1 \neq 0, \lambda_{12} \neq 0$ has the desired property in terms of giving rise to spontaneous breaking, but already at 1-loop $\lambda_2 \neq 0$. Furthermore, the full set of zeros of the $\beta$-function on the right hand side of equation (16) with non-zero $\lambda_{12}$ can be obtained explicitly. There are two solutions, either an $O(2)$ invariant model or a model with two decoupled 1-component models. Neither of these support spontaneous scale symmetry breaking.

This means that the flat direction which was essential for spontaneous breaking is lifted by quantum effects and a massless dilaton is not present.

A similar analysis holds for the model (9). In order to include all terms at leading order which are generated perturbatively, we must consider,

$$\mathscr{V} = \frac{\lambda_1}{4!}(\phi_a\phi_a)^2 + \frac{\lambda_2}{4!}\Phi^4 + \frac{\lambda_{12}}{4}\phi_a\phi_a\Phi^2. \tag{17}$$

The $\beta$-functions for the three couplings,

$$\begin{aligned}
\mu\frac{d\lambda_1}{d\mu} &= -\varepsilon\lambda_1 + \frac{3}{16\pi^2}\left(\frac{n+8}{9}\lambda_1^2 + \lambda_{12}^2\right), \\
\mu\frac{d\lambda_2}{d\mu} &= -\varepsilon\lambda_2 + \frac{3}{16\pi^2}\left(\lambda_2^2 + n\lambda_{12}^2\right), \\
\mu\frac{d\lambda_{12}}{d\mu} &= -\varepsilon\lambda_{12} + \frac{1}{16\pi^2}\lambda_{12}\left(\frac{n+2}{3}\lambda_1 + \lambda_2 + 4\lambda_{12}\right),
\end{aligned} \tag{18}$$

again lead to the conclusion that the $\lambda_2 = 0$ subspace is not invariant under the RG flow. And there are again only two solutions for the zeros of the $\beta$-functions with non-zero $\lambda_{12}$, namely the $O(n+1)$ invariant model and a decoupled $O(n)$ and 1-component model, neither of which leads to spontaneous scale symmetry breaking.

It is easy to see that the same reasoning applies to model (6) too.

# 4   Generic scalar CFT in $4-\varepsilon$ dimensions

The conclusions about the three examples also hold generally and is our main result. Either the scalar potential allows for an IR fixed point if all vevs are zero or the scalar potential has a flat direction allowing for a massless dilaton if a vev is non-zero, but not both.

Let us start with a generic tree-level potential (2) for an $N$-component real scalar field with real couplings in $4 - \varepsilon$ dimensions and assume the two ingredients necessary for our desired construction, an IR fixed point for the couplings and a flat direction. The 1-loop $\beta$-function for the couplings $\lambda_{abcd}$ is,

$$\beta_{abcd} = -\varepsilon \lambda_{abcd} + \frac{1}{16\pi^2} \left( \lambda_{abef} \lambda_{efcd} + \lambda_{acef} \lambda_{efbd} + \lambda_{adef} \lambda_{efbc} \right). \tag{19}$$

IR fixed points are given by its zeros, namely,

$$\varepsilon \lambda_{abcd} = \frac{1}{16\pi^2} \left( \lambda_{abef} \lambda_{efcd} + \lambda_{acef} \lambda_{efbd} + \lambda_{adef} \lambda_{efbc} \right), \tag{20}$$

and even though these are seemingly simple quadratic equations, a general classification of its solutions for arbitrary $N$ is still lacking [53, 54]. In any case the solutions are $\lambda_{abcd} = O(\varepsilon)$. Let us assume the flat direction, necessary for spontaneous scale symmetry breaking, is given by some non-zero vev, $\phi_a = v_a$,

$$\lambda_{abcd} v_a v_b v_c v_d = 0. \tag{21}$$

Using (20) this means,

$$\lambda_{abef} \lambda_{efcd} v_a v_b v_c v_d = 0, \tag{22}$$

which implies $\lambda_{abef} v_a v_b = 0$ for real fields and real couplings. Again using (20) we obtain,

$$\lambda_{abcd} v_a = 0. \tag{23}$$

If the field $\phi_a$ is decomposed into parallel and orthogonal components to $v_a$ then (23) means that $\mathcal{V}(\phi)$ can not depend on the parallel components at all. Which means that the $N$-component model decouples into a free massless 1-component model and $N-1$-components with a quartic potential. Continuing the argument down to $N = 1$ we conclude that the only possibility is $\mathcal{V}(\phi) = 0$ i.e. no interaction and no non-trivial IR fixed point, only $N$ independent free massless scalars.

This is the main result of our paper: spontaneous scale symmetry breaking and hence a massless dilaton can not arise in $4 - \varepsilon$ dimensional scalar CFT.

# 5 Generic scalar CFT in $3 - \varepsilon$ dimensions

It is possible to generalize the previous section to $3 - \varepsilon$ dimensions and $\phi^6$ theory. The potential in this case is,

$$\mathcal{V}(\phi) = \frac{\lambda_{abcdef}}{6!} \phi_a \phi_b \phi_c \phi_d \phi_e \phi_f, \tag{24}$$

with a totally symmetric coupling $\lambda_{abcdef}$. The leading order 2-loop $\beta$-function for the couplings is well-known,

$$\beta_{abcdef} = -\varepsilon \lambda_{abcdef} + \frac{1}{96\pi^2} \left( \lambda_{abcghi} \lambda_{defghi} + (9 \text{ permutations}) \right), \tag{25}$$

where the 9 permutations restore total symmetry in the $abcdef$ indices. Wilson-Fisher type fixed points exist and are solutions of

$$\varepsilon \lambda_{abcdef} = \frac{1}{96\pi^2} \left( \lambda_{abcghi} \lambda_{defghi} + (9 \text{ permutations}) \right), \tag{26}$$

hence $\lambda_{abcdef} = O(\varepsilon)$. See [53] for details on explicit solutions and higher loop corrections.

Now let us assume the potential (24) has a flat direction and for notational simplicity let this be $v_a = (0, \ldots, 0, v)$. Then $\mathcal{V}(v) = 0$ and using (26),

$$0 = \varepsilon \lambda_{NNNNNN} = \frac{10}{96\pi^2} \lambda_{NNNghi} \lambda_{NNNghi}, \tag{27}$$

which means $\lambda_{NNNghi} = 0$. Let us denote by $\alpha, \beta, \gamma, \ldots = 1, \ldots, N-1$ the directions orthogonal to the flat direction. In particular we have $\lambda_{\alpha\beta\gamma NNN} = 0$ and also $\lambda_{\alpha\beta NNNN} = 0$. Again using (26) we get,

$$0 = \varepsilon \lambda_{\alpha\beta NNNN} = \frac{1}{16\pi^2} \lambda_{\alpha NNghi} \lambda_{\beta NNghi}, \tag{28}$$

which leads to $\lambda_{\alpha\beta\gamma\delta NN} = 0$ and also in the same fashion as above,

$$
\begin{aligned}
0 &= \varepsilon \lambda_{\alpha\beta\gamma\delta NN} \\
&= \frac{1}{48\pi^2} \left( \lambda_{\alpha\beta\zeta\eta\kappa N} \lambda_{\gamma\delta\zeta\eta\kappa N} + \lambda_{\alpha\gamma\zeta\eta\kappa N} \lambda_{\beta\delta\zeta\eta\kappa N} + \lambda_{\alpha\delta\zeta\eta\kappa N} \lambda_{\gamma\beta\zeta\eta\kappa N} \right).
\end{aligned}
\tag{29}
$$

This then implies $\lambda_{\alpha\beta\gamma\delta\zeta N} = 0$ which is to say that the potential does not depend on the $N^{th}$ component at all and describes $N-1$ interacting scalar fields and one decoupled free massless field. Just as in the $4-\varepsilon$ dimensional case the argument can be repeated down to $N = 1$, hence a flat direction is not compatible with a non-trivial fixed point.

## 6 Conclusion and outlook

The dynamical appearance of a dilaton and its detailed properties are a non-trivial problem in QFT. An appealing playground would be a concrete top-down QFT construction which is perturbative, non-supersymmetric, renormalizable and unitary, beyond of course the prerequisite spontaneous breaking of scale invariance itself. In order to have a massless dilaton, scale symmetry should not be broken anomalously, i.e. the starting point should be a CFT. All 4 ingredients are important: the perturbative nature of the construction would ensure that all properties can reliably be calculated, the non-supersymmtric requirement would guarantee that the construction is generic enough, renormaliability would ensure that the low energy effective theory describing the dilaton and potentially other Goldstones is UV complete, and unitarity would make sure that the construction has a well-defined Hamiltonian version.

It might seem at first that fullfilling all 4 requirements would not be difficult, but actually there is no known example, even if scale symmetry is allowed to be broken anomalously, in 4 dimensions. Anomalous scale symmetry breaking, as is the case in 4 dimensional scalar QFT, complicates the identification of a potentially light dilaton. A rigorous starting point is an exact CFT, in which case an exactly massless dilaton would emerge provided spontaneous scale symmetry breaking does take place. Hence in this paper $4-\varepsilon$ dimensional scalar QFT was considered with couplings which are at an IR fixed point point of the 1-loop $\beta$-function. In principle having more than one scalar field components could allow for a flat direction for the potential leading to spontaneous breaking only, but it turns out there is no potential with the two properties simultaneously, namely an IR fixed point for the couplings and a flat direction as well.

Interestingly, in 4 dimensions, the difficulty lies in fullfilling all 4 requirements simultaneously. If models are allowed to be supersymmetric, examples do exist, most notably $\mathcal{N} = 2$ or $\mathcal{N} = 4$ SUSY Yang-Mills theories [42]. If renormalizability is dropped, one may choose specific models from a large class of effective theories; see [55–57] for recent developments.

If unitarity is dropped, again examples are known in the form of fishnet CFT's [44,49]. If the interaction is allowed to be strong, beyond the realm of perturbation theory, much less is known rigorously but one may argue that gauge theories with sufficiently large fermion content could serve as examples, at least for an approximately massless dilaton. Although in the gauge theory setup the possibility of an exactly massless dilaton was recently investigated [58].

The gauge theory situation was part of the motivation for our study. Even if it is accepted that an approximately massless dilaton is present in the spectrum, it is not at all clear what the effective theory is describing the coupled system of Goldstone bosons from chiral symmetry breaking and the dilaton. Consequently it is not at all clear what the various couplings are and what the detailed properties of the dilaton itself, e.g. its potential, is. A generic top-down model with calculable properties would probably have shed some light on some of these details.

It is still possible that non-supersymmetric, renormalizable, unitary and perturbative theories do exist with spontaneous scale symmetry breaking in 4 or $4-\varepsilon$ dimensions. One certainly needs to look beyond purely scalar QFT's and we hope to address larger classes of models in the future. It may also be the case that such theories do not exist, in which case a general proof would be desirable.

The arguments about the lack of spontaneous scale symmetry breaking in scalar QFT in $4-\varepsilon$ dimensions can easily be applied to $3-\varepsilon$ dimensions and $\phi^6$ theory as well. The results are the same, either the potential has a flat direction or its couplings are at a small $O(\varepsilon)$ IR fixed point, but not both.

In principle one could investigate the same question in $6-\varepsilon$ dimensions and $\phi^3$ theory, but in this case the potential is not bounded from below and a stable vacuum can not be defined, hence the physical meaning of a flat direction, even if exists, is questionable. Note that also from a purely CFT point of view there appears to be a qualitative difference between the $3-\varepsilon$ and $4-\varepsilon$ dimensional cases on the one hand and the $6-\varepsilon$ dimensional case on the other [59].

## Acknowledgments

DN would like to acknowledge very illuminating conversations with Csaba Csaky, Gergely Fejos, Sandor Katz, Agostino Patella, Slava Rychkov and Zsolt Szep.

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
