# Peer review of "Dilaton in scalar QFT: a no-go theorem in 4-epsilon and 3-epsilon dimensions"

_SciPost Physics, doi:SciPost Phys. 12, 169 (2022)_

## Round 1 · Referee Report · Andreas Stergiou (Referee 2) · 2022-3-11

Strengths

1- Good presentation 2- Pedagogical treatment 3- Solid and simple proof in section 4

Weaknesses

1- Work may not contain enough new results to warrant publication in current state

Report

This manuscript proves that conformal fixed points in scalar models in $d=4-\varepsilon$ have potentials that do not allow for spontaneous scale symmetry breaking (SSSB). The authors begin with a discussion of classical potentials that display SSSB and proceed to show that when quantum mechanical corrections are taken into account, the generated couplings under renormalization are such that SSSB is not possible. The proof provided is simple and mathematically correct.

This manuscript deserves publication as it adds to the knowledge we have about the behavior of conformal field theories in the $\varepsilon$ expansion below $d=4$. However, one could argue that the result presented may not be enough on its own to warrant publication.

My suggestion in order to ameliorate this concern is for the authors to also discuss the case of scalar models in $d=3-\varepsilon$, where the scalar potential involves the sixth power of the scalar field. In that case, the authors' argument in section 4 does not appear to go through in its current form, although there might be refinements. The case of $d=6-\varepsilon$ could also be discussed, where the scalar potential is cubic in the field and again the authors' argument does not appear sufficient to prove a statement similar to $d=4-\varepsilon$ in its current form. These cases could be discussed briefly in the conclusion and outlook section.

Requested changes

1- Discuss $d=3-\varepsilon$ case 2- Discuss $d=6-\varepsilon$ case

---

## Round 1 · Referee Report · Anonymous (Referee 1) · 2022-3-25

Strengths

The arguments given are elementary and are only applied at one loop. I dare say the results about the absence of a dilaton in purely scalar theories in the epsilon expansion are correct but are hardly conclusive about the absence more generally in non supersymmetric theories.

Weaknesses

The main argument in section 4 is almost trivial and rather transcends the examples given in previous sections.

Report

This paper makes a rather minor point but perhaps this has not been made explicitly before and so to this extent the paper has originality.

Requested changes

Are there examples in the literature where dilatons are known in non unitary theories? Are such examples easy to construct? Perhaps a few comments in this direction might be added.

---

## Round 3 · Referee Report · Andreas Stergiou (Referee 2) · 2022-4-14

Report

The authors' additions strengthen their manuscript significantly. This manuscript is well-written and contains enough new results to be published in SciPost Physics. I thus recommend publication.

---

## Round 3 · Referee Report · Anonymous (Referee 1) · 2022-4-20

Strengths

The arguments have improved

Weaknesses

It is still only applicable to scalar theories.

Report

The paper is short and has some remarks not seen elsewhere so I recommend publication

---

## Round 3 · Author Response

We would like to thank the referees for their comments and remarks. Following the suggestions of one of the referees we have added a new section on the 3-epsilon dimensional case where the interaction is phi^6. The same result applies as in 4-epsilon dimensions with phi^4 interaction. The 6-epsilon dimensional case with phi^3 interaction is briefly commented on.

---

## Round 3 · List of Changes

The title of the paper is extended and now is

"Dilaton in scalar QFT: a no-go theorem in 4-epsilon and 3-epsilon dimensions"

The abstract got an additional sentence "The arguments apply to phi^6 theory in 3-epsilon dimensions as well." too.

In section 1 an additional paragraph plus an additional sentence was added on the 3-epsilon phi^6 case.

Section 5 is new and is dedicated to the 3-epsilon dimensional phi^6 case.

The conclusion section (Section 6) is extended by 2 paragraphs, one about the 3-epsilon dimensional phi^6 case and a brief mention of the 6-epsilon dimensional phi^3 case.

We would like to thank the referee again for bringing up the other dimensions for discussion, we feel the additional results on phi^6 in 3-epsilon dimensions was a worthwhile addition.

The other referee had the following questions:

"Are there examples in the literature where dilatons are known in non unitary theories? Are such examples easy to construct? Perhaps a few comments in this direction might be added."

Fishnet CFT's are examples of non-unitary theories with a dilaton, which was already discussed in the introduction (Section 1). This construction can be thought of as the large-N limit of gamma-deformed N=4 SUSY Yang-Mills theory at strong deformation and weak coupling. The resulting theory is renormalizable and non-unitary and contains a dilaton. The discussion is also present in the Conclusion section.

We believe the addtions and modifications improved our paper and hence would like to thank both referees for their input.

In order to ease the next round of review of our manuscript we highlighted the added text in red.

---

## Editorial Decision

published